# Interspecific comparison of gene expression profiles using machine learning

**Artem S. Kasianov**[1], **Anna V. Klepikova**[1], **Alexey V. Mayorov**[1], **Gleb S. Buzanov**[2], **Maria D. Logacheva**[1,3], **Aleksey A. Penin**[1]*

**1** Institute for Information Transmission Problems of the Russian Academy of Sciences, Moscow, Russia, **2** Moscow Institute of Physics and Technology, Moscow, Russia, **3** Skolkovo Institute of Science and Technology, Moscow, Russia

☯ These authors contributed equally to this work.
* alekseypenin@gmail.com

**Data Availability Statement:** All relevant data are within the manuscript and its Supporting Information files and on github page of ISEEML project: https://github.com/ArtemKasianov/ICML3.

## Abstract

Interspecific gene comparisons are the keystones for many areas of biological research and are especially important for the translation of knowledge from model organisms to economically important species. Currently they are hampered by the low resolution of methods based on sequence analysis and by the complex evolutionary history of eukaryotic genes. This is especially critical for plants, whose genomes are shaped by multiple whole genome duplications and subsequent gene loss. This requires the development of new methods for comparing the functions of genes in different species. Here, we report ISEEML (**I**nterspecific **S**imilarity of **E**xpression **E**valuated using **M**achine **L**earning)–a novel machine learning-based algorithm for interspecific gene classification. In contrast to previous studies focused on sequence similarity, our algorithm focuses on functional similarity inferred from the comparison of gene expression profiles. We propose novel metrics for expression pattern similarity–expression score (ES)–that is suitable for species with differing morphologies. As a proof of concept, we compare detailed transcriptome maps of *Arabidopsis thaliana*, the model species, *Zea mays* (maize) and *Fagopyrum esculentum* (common buckwheat), which are species that represent distant clades within flowering plants. The classifier resulted in an AUC of 0.91; under the ES threshold of 0.5, the specificity was 94%, and sensitivity was 72%.

## Author summary

Interspecific gene comparisons are keystone for many areas of biological research, being especially important for the translation of knowledge from model organisms to economically important species. Currently, they are based on the concept of orthology–the orthologs are assumed to have similar functions (the so called Ortholog Conjecture). This approach is problematic for two reasons: 1) the universal applicability of Ortholog Conjecture is arguable 2) the accuracy of orthology inference is complicated due to multiple whole genome duplications and subsequent gene loss–the typical processes for most eukaryotic organisms. We report a novel machine-learning-based algorithm for the

**Funding:** This work was supported by the Russian Science Foundation, project #17-14-01315 – conceptualization (to ASK, AVK, MDL, AAP), by the Institute for Information Transmission Problems (Laboratory of Plant Genomics), project # FFNU-2022-0037 – development of ISEEML tool (to ASK, AVK, AVM, AAP), and by the Ministry of Science and Higher Education, project #075-15-2021-1064 - data analysis (to ASK, AVK, MDL, AAP). The funders had no role in study design, data collection and analysis, decision to publish, or preparation of the manuscript.

**Competing interests:** The authors have declared that no competing interests exist.

interspecific gene comparison. In contrast to previous studies, which focus on sequence similarity, it focuses on the similarity of function at the organismic level approximated by the expression patterns. As source of information for the classification, we use detailed gene expression maps. Our study for the first time proposes a metrics for comparison of expression maps suitable for species with differing morphologies and/or developmental rates. Without this, the comparisons of expression maps were possible only either for closely related species with similar morphology or for very low resolution maps. In contrast, our approach is suitable for the wide range of organisms with no limitations of their morphology and the resolution of expression maps.

This is a *PLOS Computational Biology* Methods paper.

## Introduction

Despite great progress in genome sequencing, functional characterization of genes is lagging. Most functional studies are carried out on model species, such as mice, the fruit fly, yeasts or thale cress, in the case of plants. For most other species, annotation is transferred from model organisms guided by the assumption that orthologs (genes derived from a common ancestor by a speciation event) have similar functions [1]. Thus, comparative genomic analyses are focused on the identification of orthologs [2]. The main source of information for the detection of orthologs are gene sequences (sometimes complemented by non-sequence data, e.g., gene order). Regardless of the software used to identify orthologs [3], parameters that result in a high number of true positive findings and a low number or false positives have not been identified.

This problem is most acute for plant science, because the main pattern of plant genome evolution is polyploidization followed by loss or different modes of retention of duplicated genes (for review see [4]). Lineage-specific duplications of certain groups of genes (especially ones associated with stress response and signaling systems) are also very widespread [5]. Moreover, though plants are an iconic example, many other groups of eukaryotes are also prone to gene family expansions due to either whole genome duplications or segmental duplications [6,7]. These processes lead to highly complex gene families and make the identification of a one single ortholog technically and conceptually impossible. Instead, sequence-based gene classifications divide genes into groups of co-orthologs (orthogroups). The further inference of information about the co-orthologs, in particular, the functional correspondence between them is challenging. This calls for the new types of data and approaches for interspecific gene comparisons in the context of functional genomics. Gene function and expression are tightly linked and, despite several limitations (mobile RNA, posttranscriptional and posttranslational regulation), gene expression can serve as a proxy for function. In particular, it has been shown experimentally for several model species that for duplicated genes the divergence/similarity of expression patterns is a predictor of functional divergence/similarity [8,9]. Thus, detailed gene expression maps can serve as a source of information for these novel approaches. The most widely used method for the inference of functional information from expression data is analysis of (co)expression networks (see [10] and [11]). Such networks were initially based on microarray data [12,13], but in recent years this method was replaced by RNA-seq [14,15]. These methods have provided many important insights, such as identification of a nitrogen-regulated transcription factor whose function is conserved between *Arabidopsis* and rice [16].

However, cases of interspecific analysis require a wealth of available functional information for both compared species (for a review of the procedure see [10,11]).

Few studies have attempted to classify genes based on expression data and to identify so-called functionally corresponding genes between species. One key study proposed the concept of "expressologs", which are genes that have structural similarity and similar expression patterns [17]. This study focused on ranking genes by the similarity of their expression profiles according to the following procedure. Spearman's correlation coefficient was calculated for all interspecific pairs of genes that shared structural similarity (i.e., were members of the orthogroup) and the best scoring pairs are termed expressologs. Despite its conceptual value, this approach has several limitations. First, it requires one-to-one matching of samples, which is not feasible in cases of samples with different morphologies and/or developmental rates. Second, it only allows identification of 1-to-1 interspecific pairs, whereas in fact the patterns of functional correspondence can be more complicated (e.g., two co-orthologs in one species that retain the same function as their single ortholog in the other species).

Currently, high-resolution expression maps can be easily constructed for many species using RNA-seq due to the dramatically decreased sequencing cost. Two widely used methods for the assessment of expression pattern divergence are Pearson's correlation and Euclidean distance [18–20]. However, both methods are suitable only for cases in which the compared species have highly similar morphologies and developmental rates, which is not the case for most "model species–economically important species" pairs. The importance of interspecific comparisons for efficient translation of basic knowledge into practice calls for the development of novel instruments to assess the similarity/divergence of expression profiles between species. Here, we propose a method for interspecific gene comparison to help identify functionally corresponding genes between species. This method is based on machine learning and uses two types of data (high-resolution expression data and sequence similarity). The method provides a framework for integration of functional data (transcriptomic, but the algorithm potentially can be applied to Ribo-seq data, proteomic data, etc.) with primary sequence data. This process provides a complete view of the functional correspondence between genes from different species and has potential to increase the efficiency of prediction of gene function in non-model species.

## Results

### Conceptualization

The basic assumption that underlies our approach is that the orthologs usually retain similar function (here and further, we mean function at the level of organism) in the course of evolution and their expression profiles have high similarity. Thus, if we consider a gene in one species (for example, the model plant *Arabidopsis thaliana*) and group of its co-orthologs in other species, in order to find within these co-orthologs the gene(s) that have the same function as *Arabidopsis* gene we need to find the gene(s) that have the expression pattern which is the most similar to that *Arabidopsis* gene. In order to be efficient this approach requires the high-resolution RNA-seq expression maps; they are available for a number of species (for review see e.g. [21]). From the mathematical point of view expression map is matrix of size $\mathbf{K}$ x $\mathbf{m}$, where $\mathbf{K}$ is the number of genes and $\mathbf{m}$ is the number of samples and the elements of the matrix are read counts of a gene $\mathbf{K_i}$ in the sample $\mathbf{m_j}$. For the initial development of a pipeline we used a high-resolution transcriptome map of two plants–the model species *A. thaliana* [22] and a crop plant *Fagopyrum esculentum* (common buckwheat, [23]). The maps contain 79 and 54 samples correspondingly, each in two biological replicates. These plants represent two large and distantly related groups of flowering plants (rosids and caryophyllids correspondingly)

and greatly differ in their morphology and developmental rates and are thus optimally suitable for the demonstration of the power of our approach. These two species and corresponding transcriptome maps are at the center of our study; we also performed several tests on other species and other datasets, in order to estimate the dependence on the library selection and evolutionary distance between compared species.

## Construction of ISEEML pipeline: calculation of expression scores and fractionation of orthogroups

At a first stage we performed orthology analysis for the species being compared (*Arabidopsis* and buckwheat). This was done using Orthofinder 2.4.0 [24] and revealed 4 486 orthopairs and 5 890 orthogroups (groups of co-orthologs) that include genes from both buckwheat and *Arabidopsis* (S1 and S2 Tables).

After the partitioning of the gene sets into orthogroups we proceeded with the development of the classifier (hereafter called ISEEML, **I**nterspecific **S**imilarity of **E**xpression **E**valuated using **M**achine **L**earning). This was done using a recently developed group of algorithms based on machine learning that were highly efficient in the classification of biological data [for review see [25], in particular, XGBoost, which was successfully used for expression data [26,27]. It uses known negative and positive training sets; the output is the binary classifier, which classifies the elements of the analysed set into either the positive or negative set. We used orthopairs as the positive training set, because orthologs tend to have similar expression profiles [28–30] (also see examples on S1 Fig), whereas random pairs were used as the negative set. The input for the program was the set of expression levels (read counts) for each pair of genes from two species. They are represented in a form of concatenation of vectors that correspond to the pair of genes. First vector is the string from the expression map matrix of the first species, which corresponds to the first gene within a pair, and a second vector is the string from the expression map matrix of the second species which corresponds to the second gene within a pair.

Based on these data, XGboost constructs the model for binary classification (see the Methods section "Calculation of the expression distance using machine learning" for details). The value characterizing the similarity of the expression profiles (expression score, ES) which ranges from 0 to 1, is assigned to each interspecific pair of genes. If ES>0.5 for a given pair, this pair is more similar to the orthopairs (positive training set) than to the random pairs (negative training set), and vice versa. ES values close to 0.5 are unstable and can be subliminal or supraliminal due to random factors. One of the problematic points with using orthopairs and random pairs for training is that in this case both positive and negative sets are not perfect: in the negative set there could be pairs with high similarity of expression (co-expressed genes) and vice versa, in the positive set the orthopairs with dramatically different expression patterns could be present. This calls for a special optimization for their use in training.

The inclusion of random pairs with similar expression profiles into negative set (which is unavoidable because some random pairs can be coexpressed) can influence the results of the training. In order to prevent this we performed 100 independent iterations of the training and ES calculations. A final ES value for a gene pair is the median of all values for this pair generated in the 100 iterations. The use of the classifier to calculate the ES values for the elements of the negative and positive training sets themselves is impossible, because it is based on these sets for training, and their inclusion into the analysed set will lead to biased results. To overcome this limitation, we developed a re-classification procedure using an approach of k-fold cross-validation. Briefly, the pairs from the positive and negative sets were divided into 10 subsets; nine of the 10 subsets were used as a training set, and one subset was used as an analysis

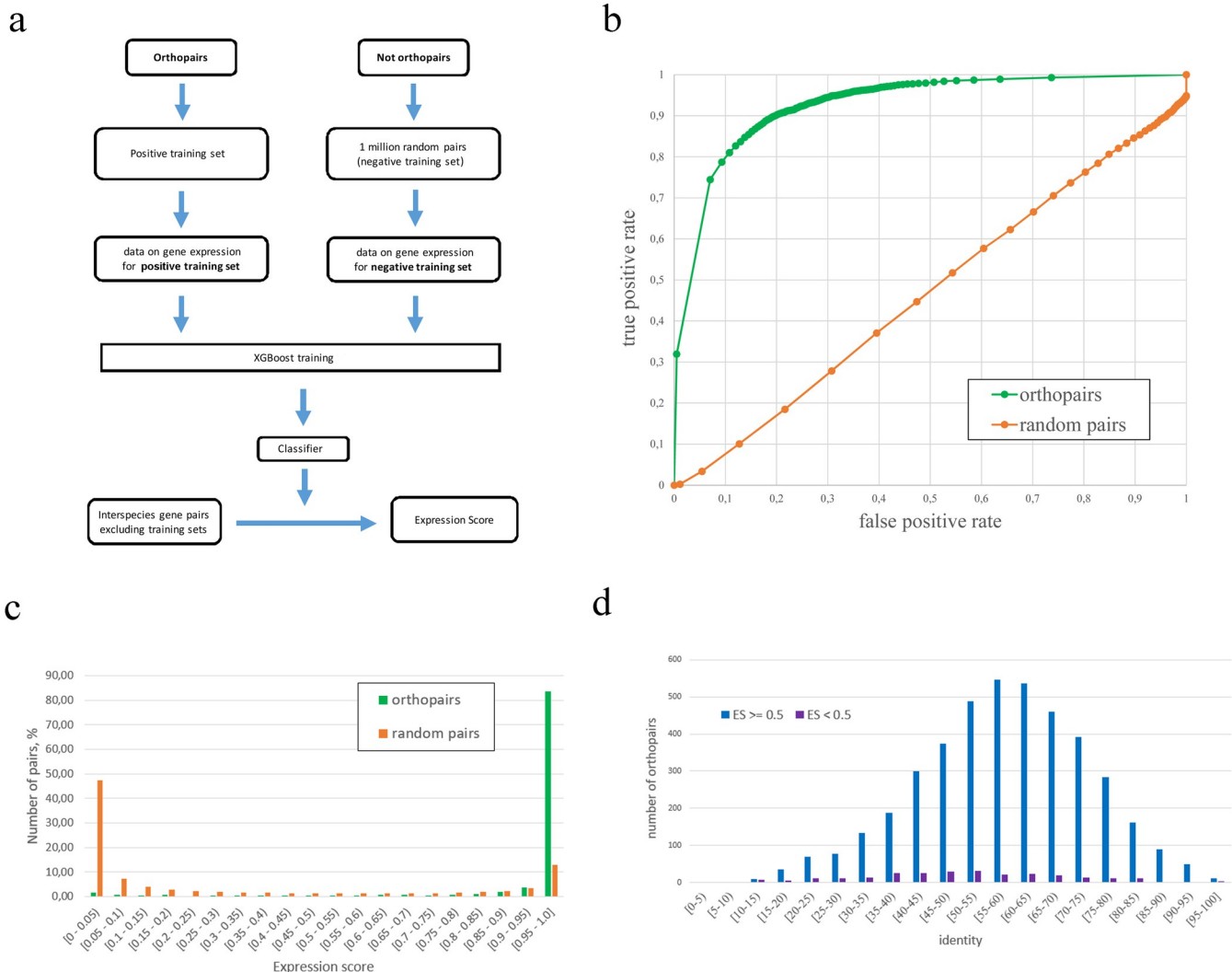

**Fig 1. Concept of the ISEEML classifier and its main characteristics.** (a) Simplified overview of the ISEEML core algorithm. (b) ROC curves for the classifier based on training with orthopairs as the positive set and random pairs as the negative set (green graph) and for the classifier based on random pairs as the positive set and another set of random pairs as negative set. (c) Distribution of the XGBoost Expression Score for the orthopairs and random pairs. (d) Distribution of identities (calculated based on Needleman-Wunsch alignment) for orthopairs with ES ≥ 0,5 and ES <0.5.

set. The procedure was repeated nine times. The overview of the core ISEEML procedure is shown in Fig 1A, and the complete flowchart is provided in S2 Fig.

The classifier described above resulted in an AUC of 0.91; under the ES threshold of 0.5, the specificity (defined as TN/TN+FP, where TN is true negative and FP is false positive) was 94%, and sensitivity was 72% (defined as TP/TP+FN, where TP is true positive, FN is false negative). In contrast, when random pairs were used as a positive training set, the classifier had an AUC close to 0.5 (Fig 1B).

Despite the high efficiency of the classifier, ~6% of gene pairs from the positive training set had an ES (calculated using the re-classification procedure described above) much less than 0.5 (i.e., they were not identified as belonging to the positive set) (Fig 1C). At the same time, their identities were not lower than those of the pairs identified as members of the positive set under reclassification procedure (see Fig 1D). Closer examination of the expression patterns of

genes from such pairs demonstrated that they indeed were dramatically different (e.g., anthers in *A. thaliana* and roots and leaves in *F. esculentum*) (for an example, see S3 Fig). Such gene pairs generate noise in the positive training set. Therefore, the estimate of the sensitivity of the method is understated. The re-classification procedure reveals these pairs; however, re-training of the model using only orthopairs that have passed re-classification is impractical because of the risk of overfitting. From a functional perspective, these pairs represent a case in which orthologs have changed their biological function or the place/condition where this function is realized. Despite the common assumption that orthologs have conserved functions, experimental data show that this scenario is not always the case [31,32]. Thus, the ISEEML allows detection of such functionally diverged orthologs. Vice versa, some genes pairs that have ES similar to that of 1-to-1 orthologs do not belong to a positive set. Indeed, these genes had very similar expression profiles but are not either 1-to-1 orthologs or members of one orthogroup (coexpressed genes) (for an example, see S4 Fig). The concluding stage of the ISEEML pipeline is the fractionation of orthogroups according expression score. From the functional point of view, if we assume that the expression pattern is linked with the function, this means the separation of genes that are functionally similar between species from ones that have different functions.

In order to do this we represent the orthogroups as a bipartite graph, with genes representing the nodes of the graph and expression scores being the weights of the edges. Edges with a weight below the threshold (0.5 by default) were removed. As a result, the graph is split into several groups (expresso-groups), which represent the genes with the high similarity of expression profiles and thus presumably functionally corresponding (Fig 2A). The application of ISEEML pipeline to *Arabidopsis*-buckwheat orthogroups have showed that out of 32 470 genes which were the members of orthogroups 5 886 genes (~18%) are singletons based on the expression score. In other words, for a gene from one species there are no genes from other species which belong to the same orthogroup and have ES > 0.5 (Fig 2B). Under the assumption that the expression pattern are associated with function this implies the widespread change of function in the course of evolution. Previous studies on gene classification based on their expression patterns [9,17] were focused on finding interspecific gene pairs with the highest expression similarity (i.e., the one with the highest correlation coefficient or lowest distance) that could be identified as "expressologs". However using comparative transcriptomics it is also possible to detection the cases in which several co-orthologs had conserved (or, vice versa, divergent) expression patterns. This helps to reveal cases in which two (or more) duplicate genes with a retained ancestral function or both (or more) have changed their function. A quantitative estimate provided by ISEEML showed that 67% of co-orthologs retained gene expression profiles (ES > 0.5) in cases with two co-orthologs versus 35% in cases with three co-orthologs (Fig 2C).

Considering orthopairs, we found that among 4486 orthopairs most have ES > 0.5, congruent with the previous observations on the higher similarity of expression profiles in orthologs (e.g. [33]). At the same time we found 268 orthopairs having expression score < 0.5. Given that the expression was shown to correlate with function [9] they are presumably the ones that underwent change of function. Our method allows finding and quantification of the frequency of such changes of function.

Since our pipeline relies on the results of the ortholog detection we tested its robustness under alternative orthology inference method. To do this we constructed orthogroups using Proteinortho [34]. It revealed 6 500 orthopairs and 4 117 interspecific orthogroups. Then the procedure of training and analysis outlined in Fig 1A was run using these orthogroups. This resulted in a classifier with efficiency similar to that constructed based on orthogroups inferred by OrthoFinder (AUC = 0.93, Fig 3A and 3B). The results of the classification based on

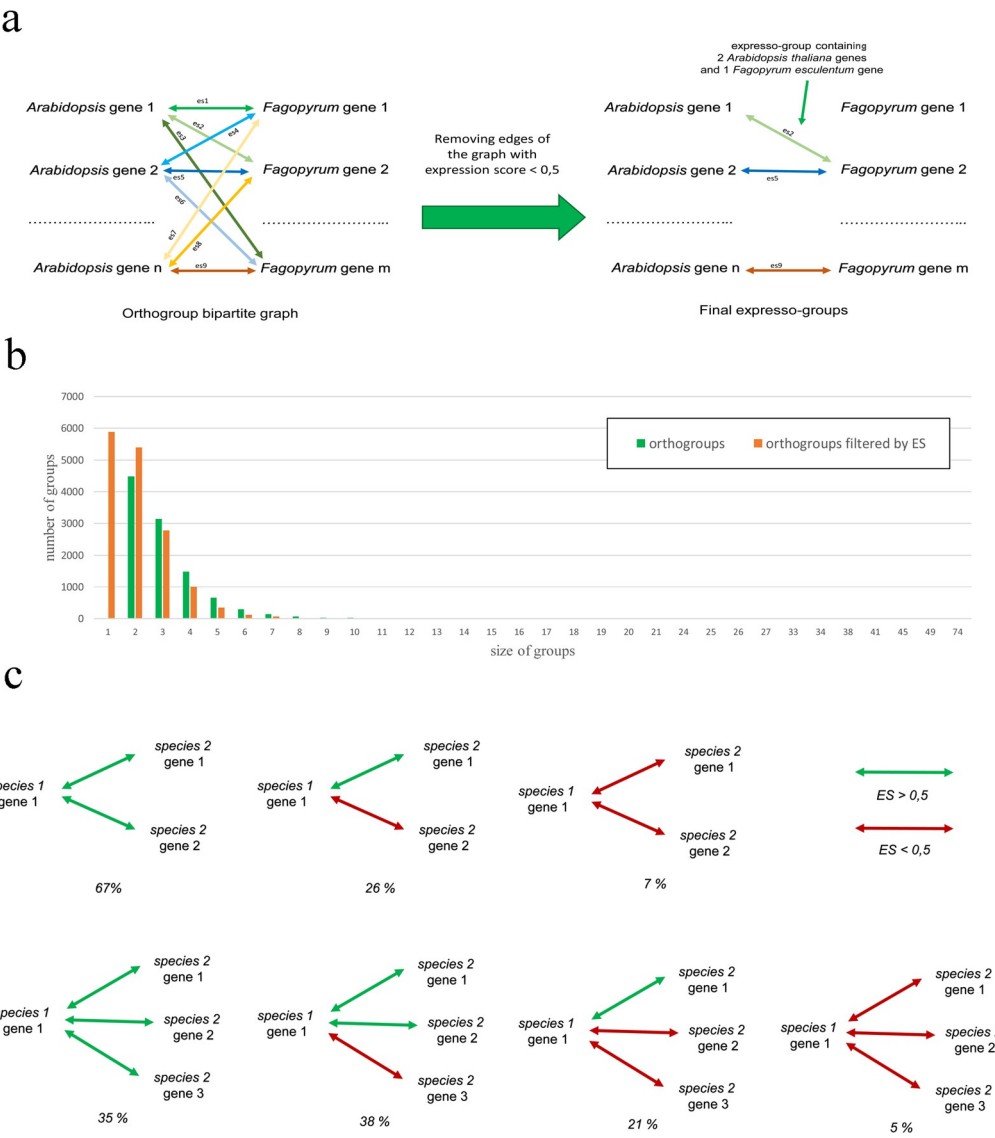

**Fig 2. Development and testing of the ISEEML pipeline.** (a) Construction of expresso-groups by fractionation of the orthogroups using ES cut-offs. (b) Distribution of sizes of the orthogroups and expresso-groups. (c) Different types of fractionation of the orthogroups and their frequencies for orthogroups with two (upper panel) and three (lower panel) co-orthologs.

Proteinortho are highly congruent with ones based on OrthoFinder (see examples for the ES > 0.5 on Fig 3C).

We also implemented the ISEEML pipeline using an alternative approach of machine learning–the one based on neural networks–that is also frequently used for the analysis of biological data, including gene expression (e.g. [35,36]). As a positive training set we employed 4 486 orthopairs inferred by OrthoFinder. We tested two negative training sets: balanced (the number of pairs in the negative set is equal to the number of pairs in the positive) and unbalanced (5-fold, the number of pairs in the negative set is five times more than the number of pairs in the positive). The advantage of using a balanced dataset is the ease of minimizing the error function. When using balanced classes, the model behaves more stable, so this method is expected to be more accurate for a given sample size. In addition, the analysis of smaller

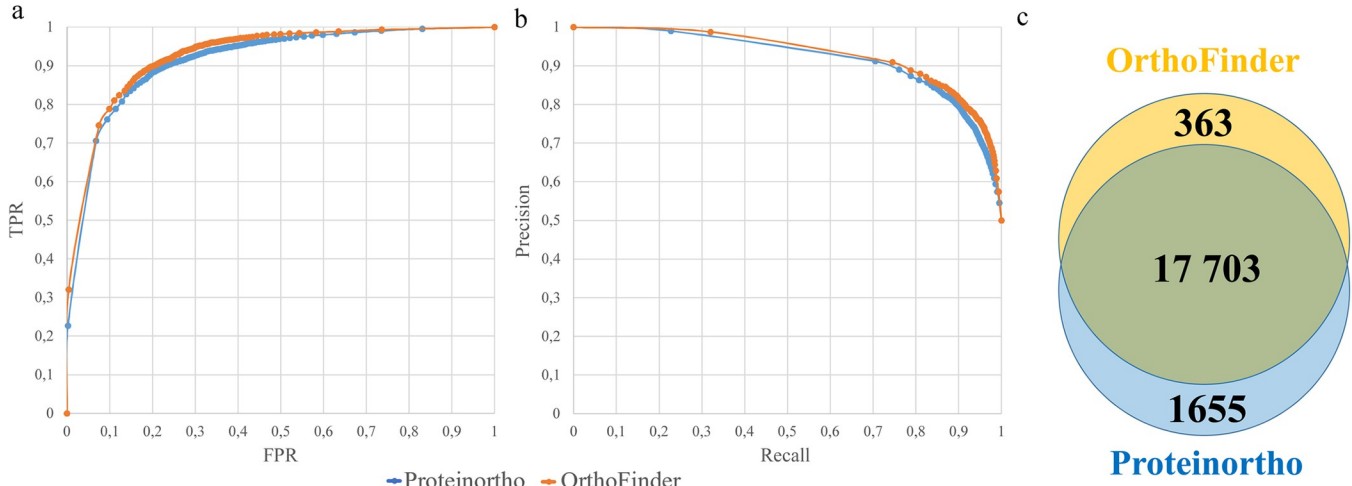

**Fig 3. Testing ISEEML efficiency under alternative methods of orthology inference.** a: ROC-curve for the classifiers based the results of Proteinortho and Orthofinder. b: Precision-recall curve for the classifiers based the results of Proteinortho and Orthofinder c: Venn diagram showing the overlap between results of the classification based on Proteinortho and Orthofinder (number denote the number of pairs with predicted ES > 0.5).

datasets is less time-consuming. However, imbalanced sets can improve the predictive power of the model. In order to avoid overfitting we performed the augmentation by addition of a random tensor with zero mean and characteristic variance of our dataset to the current batch at each epoch. This helped to improve the efficiency of a classifier (Fig 4A). The best results were obtained with unbalanced dataset (AUC = 0.94).

## Stability of ISEEML classification: species, libraries, number of samples

The results described above were obtained on two plant species that represent two large groups within the eudicots. Their estimated divergence time is 100–120 mya [37]. In order to test the

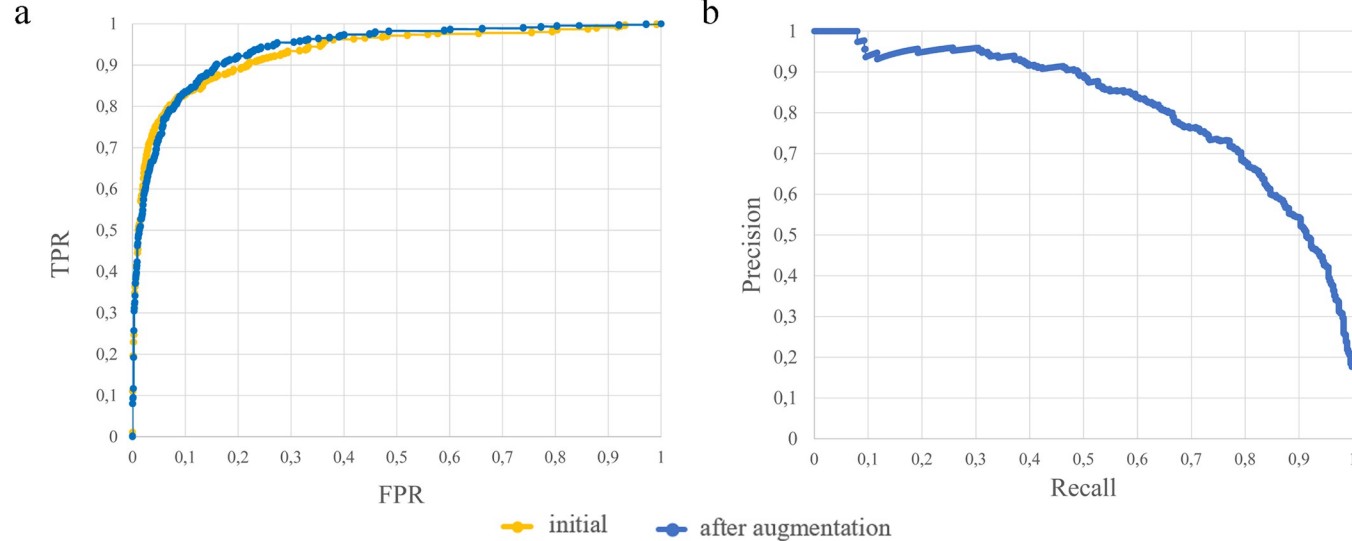

**Fig 4. Classifier based on neural networks.** a: ROC curve before (yellow) and after (blue) augmentation for the classifier based on unbalanced dataset, b: precision recall curve for the classifier based on unbalanced dataset after augmentation.

applicability of the pipeline to species with greater divergence times we analyzed two additional pairs of species: *Arabidopsis* and *Zea mays* (maize) and buckwheat and maize. In both pairs the species represent two largest groups within flowering plants–dicots and monocots correspondingly. Their estimated divergence time is 130–150 mya [38–40]. We employed data from maize transcriptome atlas [41] which encompass similar set of organs and stages as *Arabidopsis* atlas, however with focus on root samples. This also allows testing the robustness of the classification under a varying set of libraries. For buckwheat and *Arabidopsis* we used transcriptome maps collected in the same controlled conditions, at the same time of the circadian cycle, libraries prepared with the same reagents etc. This could in principle affect the efficiency of classifier (artificially increasing it due to the avoidance of noise and biases introduced by environmental conditions and technical factors). The maize transcriptome atlas is much more diverse with the regard to the growing conditions (in particular, it includes samples grown in the field) and thus one may expect higher variation in the expression profiles. We inferred orthogroups for *Arabidopsis*–maize and buckwheat–maize genes; this resulted in 4619 1-to-1 orthopairs and 5504 orthogroups for *Arabidopsis*–maize and 3336 1-to-1 orthopairs and 5905 for buckwheat–maize. The training resulted in a classifier with efficiency similar to that for *Arabidopsis*–buckwheat pair: AUC *Arabidopsis*–maize = 0.95, AUC buckwheat–maize = 0.92 (see also Fig 5A and 5B).

Since our approach aims at the classification of genes based on their expression patterns, the important characteristic that may influence the accuracy of classification is the breadth of expression pattern. In order to estimate the influence of this factor we selected genes with broad and narrow expression patterns and analyzed the success of classification for these two groups separately. The results showed that genes with broader expression patterns are classified better than ones with narrow pattern (S5 Fig). The effect is more pronounced for the comparisons that include maize because transcriptome map of maize includes set of samples which are less well matched with samples of *Arabidopsis* and buckwheat transcriptome maps than these two latter are matched between themselves (in particular maize map is focused on root development, though contains above-ground vegetative and reproductive organs as well). This is expected because if a gene is expressed in some narrow specific pattern, it may miss from the transcriptome map if the set of samples is not exhaustive.

An important question for the new method is its stability in a case with changes in the samples used for the expression analysis. To estimate the influence of sampling, we performed several tests based on the removal of samples. To select the samples for removal we used the following procedure: 1) constructed a distance tree based on expression profiles (the measure of distance is 1- Pearson correlation) 2) this tree is "cut" at different distances ranging from 0.1 to 0.9. This results in clusters where the number of samples in a cluster depends on the distance (for the least distance, 0.1, the number of clusters is maximal and for 0.9 –the largest distance–it is minimal) 3) a random sample from each cluster is retained for the analysis (see illustration for the distance 0.5 in S6 Fig and a number of samples being retained for each distance–in S3 Table) and other are removed. After the selection of samples we re-run ISEEML pipeline and compared ES values (for the pairs of genes from orthogroups) inferred from these reduced dataset with ones based on complete set. The results show high congruence of the ES for larger subsets of samples and its gradual decrease towards smaller subsets, as expected (S7 Fig).

Despite the stability of the results, the exclusion of some samples, especially ones with high number of tissue-specific genes can lead to inaccurate classification of such genes. We illustrate this with two types of samples–anthers and roots. Anthers have a specific gene expression profile that drastically differs from that of the other samples in the transcriptome maps. Roots are not photosynthetic and thus lack expression of a large fraction of the genes associated with photosynthesis, which is in contrast to above-ground axial organs, such as stems. We excluded

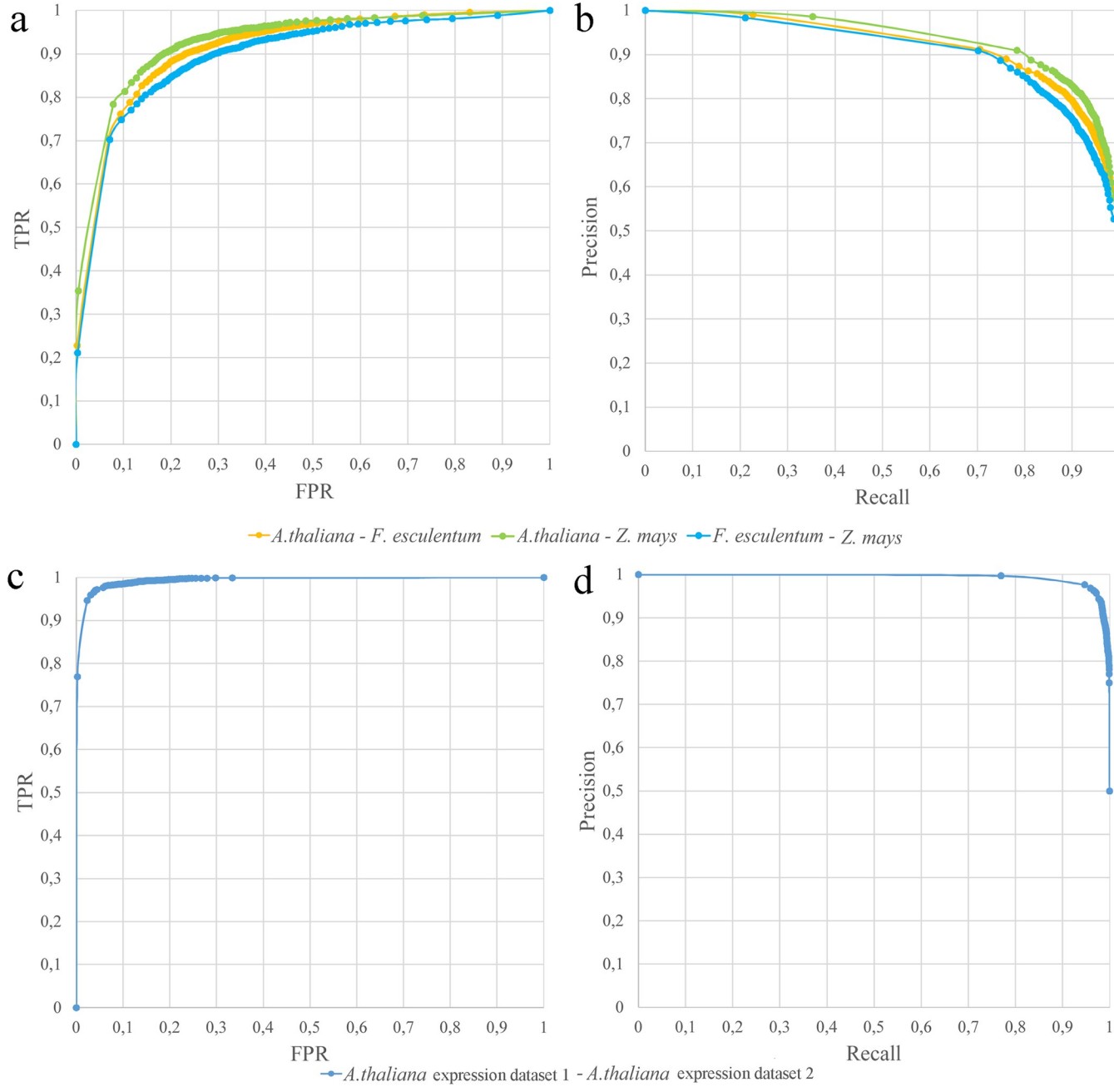

**Fig 5. Results of ISEEML classifier at different evolutionary distances.** (a) ROC curves for the ISEEML pipeline applied to three interspecific pairs: Arabidopsis–buckwheat, Arabidopsis–maize, buckwheat–maize. (b) Precision-recall curves for the same pairs. c: ROC curve for the ISEEML pipeline applied to the pair Arabidopsis-Arabidopsis where different sets of transcriptome samples were taken as a source of information on expression profiles. d: Precision-recall curve for the same input data as in c.

these samples, re-run the classification and compared the ES for genes from orthogroups with ones obtained for a complete set. Most pairs had similar ES values and were classified correctly (S8 Fig). However, for some orthopairs, the classification results changed. This finding highlights that classification based on expression requires accurate sampling that reflects all organs and developmental stages, at least for tissue-specific genes.

In order to provide the upper estimate for the efficiency of a classifier we performed a test based on comparison of *Arabidopsis* genes against themselves–the case where functional equivalence of all genes is known with 100% reliability. We took two transcriptomic datasets– the one is a transcriptome map from Klepikova et al. 2016 [22] used in above-described analyses and the other a diverse set of publicly available RNA-seq libraries (for a full list see S4 Table). As a positive training set we used 4473 pairs of *Arabidopsis* genes (each pair contained the same gene listed twice, as belonging to different species) and as negative set– 4473 random pairs. As expected, this resulted in almost perfect classification (AUC = 0.99, see Fig 5C and 5D). This also highlights that the influence of noise in gene expression data introduced by the conditions, method of library preparation and sequencing and other environmental and technical factors has little influence on the performance of ISEEML pipeline.

## Comparison with distance-based metrics of gene expression similarity

A common approach for estimation of expression similarity in groups of duplicated genes is the calculation of Euclidean distance [19,30,42–44]. Thus we compared our results described above with that of a classifier based on Euclidean distance. The use of this metric for interspecific analysis requires matching of the samples, which is not always straightforward due to differences in morphology and developmental rate. To account for this issue, we used additional optimizations based on the grouping of transcriptome samples and the search for minimal distance within groups (see Methods for details). As expected, the distribution of distances between orthopairs and random pairs was drastically different (S9 Fig) thus indicating that it has some potential for the classification. However, these distributions overlap, and, as expected the classifier based on the Euclidean distance has a moderate AUC (0.65 for *Arabidopsis*–buckwheat, 0.6 for *Arabidopsis*–maize and 0.63 for buckwheat–maize, see S10 Fig).

## Software realization of the ISEEML pipeline

The set of scripts that constitute ISEEML pipeline is available on GitHub: https://github.com/ArtemKasianov/ICML3

The software consists of three main modules:

1. A module that provides training of models.

2. A module that predicts expression scores for pairs of genes included in the training sample.

3. A module that predicts expression scores for an arbitrary set of gene pairs.

As input data, the software uses sets of expression profiles for the two species being compared and a set of orthopairs. A full cycle of analysis on one thread per iteration takes about 10 minutes on average for species with no more than 30 thousand genes and no more than 200 samples in the expression profile (the calculations were performed on AMD EPYC 7452 processor).

## Discussion

The advances in DNA sequencing technologies have led to an enormous increase in the number of genomes and transcriptomes being characterized in the recent years. However, the functional characterization of the genes is still lagging behind. This is especially true for plants due to the multiple whole genome duplications that are the inherent part of plants' evolutionary history. The existence of plant genes as members of multigene families hampers both existing bioinformatics methods based on sequence similarity and experimental methods such as genome editing. Thus there is a growing interest in the development and application of

approaches based on additional types of data, in particular, RNA expression profiles (e.g. [9,17,42,45,46]).

A usually used approach for interspecific comparisons of gene expression profiles is based on calculation of the Euclidean distance between expression profiles. This metrics was used for several studies on animals that are either closely related or have similar body plan, in particular, for a search of neofunctionalization in mammals [20] and in *Drosophila* [19]. However, the morphology, life cycles and developmental rate in eukaryotes are highly variable and, the direct matching of the samples between species (which is necessary for the calculation of Euclidean distance) is often impossible, especially at large evolutionary distances. Even under the generally conserved body plan (like, for example, in flowering plants or mammals) this is a problem due to the great variation stemming from reductions and modifications of the organs and, in some cases, to their unclear homology (see e.g. [47,48]). Our method overcomes this limitation due to the application of machine learning algorithms that do not require matching of samples.

We expect that further developments will improve the usability of the classifier. First, expansion of the species set will greatly decrease the number of singletons. A shortcoming of this method is its sensitivity to the genome assembly and annotation quality; however, this issue is the case for virtually any gene classification method. The decrease in the sequencing cost (including third-generation technologies that generate very long reads and thus are favourable for genome assembly) and improvement of bioinformatics pipelines will minimize the influence of this factor. Another important consideration is the resolution of transcriptome maps. Apparently, the classification of tissue (organ/stage/condition)-specific genes will not be accurate if the corresponding tissue is not sampled in the map. Thus in order to get the most accurate picture of the expression similarity of the organismal level, we recommend sampling of 20–90 samples (this range is taken from several recent articles on transcriptome maps [49–51]) which necessarily include ones that have high fraction of specific genes (for example, brain and testis in mammals or anthers and roots in flowering plants). However, since the ES (as well as any other measures of coexpression, including interspecific coexpression) is a dataset-dependent measure, for the studies focusing on specific processes (e.g. stresses, aging) it is necessary to include in the comparison samples representing those processes.

The important advantage of the classifier is that it is suitable for analysis of publicly available data with sufficient organ and stage diversity. This method allows the analysis of big data in automatic or semiautomatic mode based on metadata and the main technical parameters (such as the % of mapped reads). In addition to application of the algorithm outlined here, our approach can be applied for the identification of coexpressed genes between species.

The complexity of gene families is one of the main factors hampering the translation of knowledge derived from model species to other ones, including agriculturally important species. The identification of functionally similar genes will allow efficient selection of candidate genes and as a result decrease the time and other resources required for the development of new cultivars.

## Methods

### The estimation of gene expression

As an input data we used read counts obtained by the mapping of RNA-seq data. The reads were mapped on the annotated genome using STAR [52] and counted using HTseq-count. The versions/sources of genome annotation were TAIR10 for *Arabidopsis thaliana*, the annotation from [23] for buckwheat and RefGene_v4 for maize. Main sources of the RNA-seq data were transcriptome maps for *Arabidopsis* [22], maize [41] and buckwheat [23] (the raw data

are available in SRA, accession numbers and library names are listed in S5 Table). For the self-consistency test based on a comparison *Arabidopsis–Arabidopsis* we used a set of RNA-seq samples from different studies publicly available in SRA (for a list of samples see S4 Table).

### The search for orthologs

For most analyses, we used orthogroups (including 1-to-1 orthologs) inferred by OrthoFinder 2.4.0 [24]. In order to test the sensitivity of our approach to the method of the identification of orthologs we also used Proteinortho v. 6.0.33 [34].

### Calculation of identities

In order to provide comparison between ES and identities we calculated identity using a custom script developed based on nwalign [53], which is the realization of the Needleman-Wunsch algorithm of global pairwise alignment. The following alignment parameters were used: matrix = 'BLOSUM62.txt', gap_open = -11, and gap_extend = -1

### Calculation of the expression distance using machine learning

As input for the expression profile analysis, we used the following data:

1. A set of 1-to-1 orthologs (orthopairs).

2. Expression values (non-normalized read counts) for each gene in each sample. For each interspecific gene pair, a vector of expression weights was constructed. The components of this vector are the expression levels (non-normalized read counts) for all samples of the transcriptome maps.

To assess the similarity of expression profiles for interspecific gene pairs, we used two machine learning approaches. The first one, which is central in the current study is gradient boosting of decision trees. The result of processing by the classifier of an expression score vector for a pair of genes is a value ranging from 0 to 1, known here as the "expression score" (ES). An ES<0.5 corresponds to the case in which a pair of genes has different expression profiles, and an ES>0.5 corresponds to the case in which a pair of genes has similar expression profiles. ES = 0.5 indicates that the classifier cannot decide whether the profiles are similar.

For construction of a binary classifier, negative and positive training sets are required. A positive training set is a set of pairs with the most similar expression patterns possible, and a negative training set is a set of pairs with the most different expression patterns possible. For the positive set, we used orthopairs based on following assumptions: 1) orthologs have similar functions (this assumption is known as the ortholog conjecture) and 2) a gene is functional when it is expressed 3) the expression patterns can be used as a proxy for functional patterns. As the negative set, we used randomly selected pairs. The number of such random pairs was equal to the number of the pairs of the positive set; the selection was performed from a set that did not contain the elements of the positive set (orthopairs) and without replacement. For construction of the negative set we used a custom script, GetNegativeRandomSet.pl.

Obviously, direct calculation of ES for pairs that are elements of either the positive or negative set is not correct, because they are used for training of the classifier. To overcome this limitation, we developed a special re-classification procedure. First, we divided the positive and negative training sets into 10 parts using the custom script GetRandomFoldOfOrthopairs.pl. Then, 10 models were trained; to train each model, one subset each from a negative and positive set was excluded. Then, the ES was calculated for each subset using the model trained under exclusion of this subset. After this procedure, the model using the complete training sets

was constructed and used to calculate the ES for all remaining interspecific pairs within each orthogroup. For the machine learning, we used XGBoost [54], which is a programme package that performs gradient boosting of decision trees. To convert the data from the positive and negative training sets to the format used by XGBoost, we developed the custom scripts GenerateSVMFile.expression.pl and GenerateSVMFile.expression.pairs.pl. To run the training and prediction with a model, we used the custom scripts XGBoostTree.saveModel.py and PredictByModelXGBoost.10.py. The XGBoost output is a model for binary classification (the classifier).

The optimal parameters were chosen based on one iteration. Within the iteration, the classifier quality was assessed during cross-validation. We used the AUC value as a criterion for the classification quality. The following parameters were tested:

1. subsample in the range [0.7, 1] in increments of 0.1;

2. colsample_bytree in the range [0.7, 1] in increments of 0.1; and

3. colsample_bylevel in the range [0.7, 1] in increments of 0.1.

As a result of testing, we found that the best result was achieved under the following parameters: sabsample = 1, colsample_bytree = 1, and colsample_bylevel = 1. The change in AUC was subtle and was not more than 0.02.

For these **subsample**, **colsample_bytree**, and **colsample_bylevel** values, we tested the following parameters:

1. **gamma** in the range [0, 10] in increments of 2;

2. **alpha** in the range [0, 10] in increments of 2; and

3. **lambda** in the range [0, 10] in increments of 1.

We found that the AUC value was rather stable under these parameters, because the difference was not higher than 1%. Thus, we decided to use the default parameters (alpha = 0, lambda = 1, and gamma = 0).

In summary, we used the following parameters for training (the default values for the parameters not indicated below).

1. max_depth = 0; the maximum number of trees was selected automatically depending on the estimate of training efficiency, and

2. scale_pos_weight = 1. Because the positive training is equal to negative at each iteration.

Algorithms that use randomization can generate different results from run to run. In this study, we use randomization for construction of a negative training set and generation of subsets of the positive and negative training sets. To overcome the possible adverse effects of random sampling, we ran 100 iterations of the algorithm (model training and ES calculation). Then, the ES values generated at each stage for each iteration were collected in a table using the GenerateTableWithAllReadCounts.pl script. For each pair, the final ES was calculated as a median (which is good, because the method is not sensitive to outliers) of the ES resulting from each iteration using the CountMediansForTableRows.pl script.

For *Arabidopsis*–buckwheat pair we also tested another method of machine learning, based on fully connected neural networks (NN). Fully connected NN are neural networks, each neuron of which transmits its output signal to other neurons, including itself. All input signals are provided to all neurons. The output signals of the network can be all or some of the output signals of neurons after several epochs of the network functioning. Similar to the above-described

approach based on decision trees, we used 1-to-1 orthologs as positive set and random pairs as negative. We used two negative sets of various sizes–balanced (fraction = 1, number of pairs = 4472) and unbalanced (fraction = 5, number of pairs = 22360). The calculations were performed using Torch 1.12.0 with the following parameters:

params = {hidden_layers = [200,100,50,20];

relu = 'relu';

dropout = 0.1;

eps = 1e-5;

momentum = 1e-2;

batch_size = 200;

epochs = 30;

learning_rate = 1e-3}

In order to avoid overfitting, we used augmentation by adding a random tensor with zero mean and characteristic variance of our dataset to the current batch at each epoch.

## The fractionation of the orthogroups using expression scores

The key application of our pipeline is the fractionation of the orthogroups, i.e. the finding within the orthogroup which contains 1-to-many or many-to-many co-orthologs the pairs (or larger groups) which have similarity of expression profiles and are thus plausible functionally similar. We represented the orthogroups as graphs, where edges are connecting all possible interspecific pairs. The ES value was assigned to each edge. The threshold for ES was 0.5, which was selected based on XGBoost logic. As a result of the prediction for each example, the method returns values from 0 to 1; if the value is less than 0.5, the example is considered to belong to a negative set class; if it is greater than 0.5, the example is considered to belong to a positive set class. In our study, if the ES is less than 0.5, the pair is more similar to the random pairs than to the orthopairs, whereas the opposite is true if the ES is >0.5. Edges with ES values less than 0.5 were removed from the graph. As a result, the graph was split into several connected components. The genes that correspond to the nodes in each of the components have similar expression profiles and can be joined in a group known as an expresso-group based on analogy to the orthogroup. For identification of the components, we used an algorithm based on the depth-first search realized in the script PrintOrthogroupsAfterCutByTreshold.pl.

## Calculation of Euclidean distances

In order to provide comparison with previously developed approaches for comparisions of expression profiles we calculated (pseudo-)Euclidean distance as described in Penin et al., 2019 [30]. First, for each species (*A. thaliana*, *F. esculentum*, *Z. mays*) read counts were normalized using median method from DESeq [55]. Then, the normalized read counts were incremented by 1. While sharing similarity, transcriptome maps of the species being compared varied in sample content reflecting plant morphology. Thus, the direct match of several samples was impossible and such samples were grouped (sample groups are provided in S6 Table; note, that not all samples were used in the analysis and sample sets for *A. thaliana*–*F. esculentum*, *A. thaliana*–*Z. mays*, and *F esculentum*–*Z. mays* comparisons differed). In each comparison gene read counts were divided by the median of gene expression level.

Then, the pseudo-Euclidean distance was calculated for each pair of genes in the comparison (e.g. *A. thaliana–F. esculentum*) as follows: one of the biological replicates was randomly taken for each sample of first species (e.g. *A. thaliana*) and second (e.g. *F. esculentum*);

If a sample of *A. thaliana* was in direct match with a sample of *F. esculentum*, *F. esculentum* median-normalized read counts were subtracted from *A. thaliana* median-normalized read counts, and the residual was squared.

If a group of *A. thaliana* samples was compared with a group of *F. esculentum* samples, the residuals of median-normalized read counts were counted for each pair of *A. thaliana–F. esculentum* genes. The minimal residual was squared.

All squared residuals were summed, and a square root of the sum (a pseudo-Euclidean distance) was calculated.

The steps 1–4 were repeated 100 times to evaluate the contribution of expression variance.

## Supporting information

**S1 Fig. Example of the expression profiles in orthopairs. Color intensity denotes expression level.**
(PDF)

**S2 Fig. Complete scheme of the ISEEML pipeline.**
(PDF)

**S3 Fig. Example of the drastically different expression profiles in orthopairs.** Color intensity denotes expression level.
(PDF)

**S4 Fig. Example of expression profiles for gene pairs from random pair set that have low identity but high expression score.**
(PDF)

**S5 Fig.** a. ROC curves for the classification of genes with narrow and broad expression patterns b. Precision-recall curves for the classification of genes with narrow and broad expression patterns
(PDF)

**S6 Fig. Example of the principle of sample selection in downsampling test. Horizontal red line denotes distance at which the tree was cut into clusters (0.5 in this example, varies from 0.1 to 0.9 in complete analysis).** Big squares of orange, yellow, green, blue and violet color denote clusters. Samples in red squares—samples randomly selected from each cluster.
(PDF)

**S7 Fig. Comparison of the ES for orthopairs between ones inferred from the complete set of samples and from downsampled set.**
(PDF)

**S8 Fig. Comparison of the ES for orthopairs between ones inferred from the complete set and the set where some samples were removed (left–anthers excluded, right–root excluded).**
(PDF)

**S9 Fig. Distribution of Euclidean distances in orthopairs and random pairs (panels a, c and e show the complete range of values, b, d and f–the inset showing the distance in the range from 0 to 10.**
(PDF)

**S10 Fig. ROC curves for the classifier based on Euclidean distance.**
(PDF)

**S1 Table. Orthofinder orthopairs.**
(XLSX)

**S2 Table. Orthofinder orthogroups (excluding orthopairs).**
(XLSX)

**S3 Table. The number of samples retained for the downsampling analysis under different distances.**
(XLSX)

**S4 Table. List of Arabidopsis thaliana RNA-seq samples taken for self-consistency test.**
(XLSX)

**S5 Table. List of Arabidopsis thaliana, Fagopyrum esculentum and Zea mays RNA-seq samples from transcriptome maps.**
(XLSX)

**S6 Table. List of sample groups used for the calculation of Euclidean distance.**
(XLSX)

## Acknowledgments

We thank American Journal Experts (https://www.aje.com) for editing this manuscript, Alexandra M. Kasianova and Elizaveta M. Gunko for their assistance with the testing of ISEEML pipeline in MacOS and Dmitry D. Sokoloff for helpful discussion.

## Author Contributions

**Conceptualization:** Maria D. Logacheva, Aleksey A. Penin.

**Data curation:** Artem S. Kasianov, Anna V. Klepikova.

**Formal analysis:** Artem S. Kasianov, Gleb S. Buzanov, Aleksey A. Penin.

**Funding acquisition:** Aleksey A. Penin.

**Investigation:** Aleksey A. Penin.

**Methodology:** Artem S. Kasianov, Aleksey A. Penin.

**Project administration:** Aleksey A. Penin.

**Software:** Artem S. Kasianov, Alexey V. Mayorov.

**Supervision:** Aleksey A. Penin.

**Validation:** Artem S. Kasianov, Aleksey A. Penin.

**Visualization:** Artem S. Kasianov, Aleksey A. Penin.

**Writing – original draft:** Maria D. Logacheva, Aleksey A. Penin.

**Writing – review & editing:** Maria D. Logacheva, Aleksey A. Penin.

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
