## [Decision Letter · Decision Letter 0]

21 Feb 2022

Dear Dr Penin,

Thank you very much for submitting your manuscript "Interspecies gene classifier based on the analysis of RNA-seq data using machine learning" for consideration at PLOS Computational Biology.

As with all papers reviewed by the journal, your manuscript was reviewed by members of the editorial board and by several independent reviewers. In light of the reviews (below this email), we would like to invite the resubmission of a significantly-revised version that takes into account the reviewers' comments.

We cannot make any decision about publication until we have seen the revised manuscript and your response to the reviewers' comments. Your revised manuscript is also likely to be sent to reviewers for further evaluation.

Sincerely,

Andre Kahles, PhD

Guest Editor

PLOS Computational Biology

Ilya Ioshikhes

Deputy Editor

PLOS Computational Biology

Reviewer's Responses to Questions

**Comments to the Authors:**

Reviewer #1: Kasianov et al. describe a method to find what they call "expresso-groups" of genes and apply it to a pair of plant species. At the core is a tree-based binary classifier of gene pairs into expressologs and non-expressologs. It takes the vectors of unnormalized read counts in a number of libraries for each species as input and outputs an "ES" probability. The classifier is trained to reproduce orthologous pairs found with OrthoFinder. Their program ICML can predict orthology with reasonable accuracy using not sequence but expression profiles. I see a potential high relevance in complementing orthology search and for discovering interesting cases where the expression profile changed much but the sequence not or vice versa. Very importantly, the approach works on count matrices of shape g x n and G x m, where g and G are the numbers of genes in the two species, n and m are the numbers of RNA-Seq libraries from the two species and no meta information on the libraries is required (e.g. tissue types or conditions). The main difference to related previous work is that no correspondence between libraries is required and therefore n != m is allowed.

The approach is genuine, generally applicable and of potentially broad interest. However, the manuscript requires more details and an improved readability. The differences between orthology and "expressology" have not been examined much and neither has the dependence on the library selection or evolutionary distance. The description of the ML method plays a minor role.

I found the title misleading as a gene classifier could for example find functional classes. Instead the authors present a way to learn the similarity of expression profiles of gene pairs.

"PLOS Computational Biology requires authors to make all author-generated code directly related to their study’s findings publicly available" and "authors must clearly provide detail, data, code, and software to ensure readers' ability to reproduce the models".

However, the ICML3 repository contains binary files (makemodel and predictAllByPortion) which appear to have the main functionality but it is not documented where they come from or for which platform they were compiled on. Also, several scripts mentioned in the manuscript are not in the repository, e.g. XGBoostTree.saveModel.py and PrintOrthogroupsAfterCutByTreshold.pl. Please make your source codes available. Please also state the program and version of XGboost you used and give a link to the repository.

The method is not described carefully and thoroughly. It becomes only clear very late what the authors have done, e.g. they perform a binary classification of pairs of integer-valued vectors of sizes n and m. The description should be more formal, e.g. what kind of mathematical object is an expression map? Please mention early the size of n and m in your test data. Of what shape is the input to your classification trees?

In contrast to the protein sequences, the expression profiles are random (technical variation, biological variation, choice of tissue, age and conditions, no clear correspondence of samples) and may overall be insufficient to decide which genes are corresponding functionally, e.g. when the sampled libraries are not sufficient to discriminate between functions. High variation in the predicted orthology pairs could render the orthology-finding resuls largely irreproducible.

A plausibility test in which the functional equivalence of all genes is known 100% would be to compare A. thaliana against itself. Only, the expression profiles would be calculated using different RNA-seq libraries. There are plenty of libraries is SRA to draw samples from. Further, a sample of a few pairs of species would give a better impression of the impact of the randomness that is additionaly introduced compared to a sequence based method.

ICML requires that one runs orthology-finding to obtain a training set. Moreover, if ICML achieved perfect accuracy it would render itself superfluous. It is therefore important to show that the disagreements between the ortho- and expressologs are not merely classification errors but have a biological meaning. This could for example be done for cases where one has a (partial) correspondence between libraries and can compare expression profiles with established methods.

In line 177, the authors imply that certain differences between ortho- and expresso-groups are due to neofunctionalization. This hypothesis is not supported by evidence. E.g.~it could be that a pair of samples from the two species can mostly be matched up well as they are from the same tissue, but one sample is perhaps under a certain condition that changes the expression level of a few genes only. A low ES score could merely be a misclassification error. Perhaps this error probability could be bounded with above plausibility test.

Can you please discuss why the large expresso-groups ("orthgroups") seem to include only 1 gene from one species and very many from the other species? It would suprise me if many gene families had several dozen genes of the same function in one plant but are single-copy in another plant. Could these be similarly regulated genes that have very different functions?

A discussion would be helpful on the influence you expect on the selection of samples, e.g. when you mainly have samples from xylem sap of one plant and mainly have samples from root from the other.

Have the authors tried another ML method? Why have they chosen trees?

Minor Issues:

- typo in abstract: or -> of

- ES is used in the abstract before it is defined.

- Line 120. I suggest to not use "classification" in that sense as readers of a paper with ML in the title will think of the usual meaning.

- Line 131. "set of expression levels" - is the _vector_ of expression levels meant?

- The caption of Figure 1 seems to be messed up as b) is a ROC curve, not a model.

- Line 159. "identities" - state that this refers to a NW-Alignment.

- In Figure 1e one cannot see how much the scatterplot is concentrated around the diagonal, as the dots overlap much. What is the correlation coefficient?

- Line 185. oa -> of

- Grammar error in Figure 2a

- There is neither a caption nor a reference to Figure 2d.

- Line 218. Why would the thresholding and collection of connected components not work on the basis of correlation coefficients in place of ES?

- Line 241. What is 'stream'?

- Line 246. Grammar error (plural singular mismatch).

- Wrong indentation after line 284.

- Give a reference for nwalign.

- Line 304: What are the weights you are referring to?

- The parameters in line 335 ff are not introduced.

- Line 165: From Figure 1b one cannot see the accuracy at a certain threshold. From Figure 1c) it looks like at the ES threshold of 0.5 the sensitivity should be >0.9 and the specificity is lower. Please give the formula and define what you refer to as specificity.

Reviewer #2: The authors suggest a machine learning approach to refine orthology assignment for a pair of genes based on their expression pattern. The method assigns for each candidate pair an expression score. This score reflects the similarity of the expression pattern among the group of orthologous pairs compared to that of random pairs. The idea behind this method is very nice and shown to be potentially useful for refining orthology assignment, which is, traditionally, based on sequence identity.

Major

1. My main concern is about the generality and applicability of the suggested method to other species. As the method is based solely on expression patterns, a major issue is how sensitive the results are to the quality of the data and its availability. There is a strong assumption that orthologous genes have similar expression patterns while it is unclear from this study how much it depends on:

a. The evolutionary distance of the compared species

b. The number of samples required to construct an informative and representative expression pattern

c. Should both species have expression patterns from similar tissues/organs? Same developmental stage? Similar environmental conditions?

I think that the sensitivity of the results should be rigorously tested also given these issues. For example, the authors show that the correlation scores with and without specific samples are still high (but do not quantify it). I would also like to see some analyses demonstrating, for example, how sensitive are the results to the number of samples used. In the current manuscript, the authors only state one sentence about this issue: “This finding shows that classification based on expression requires accurate sampling that reflects all organs and developmental stages.” I think these analyses should be expanded. In addition, Could the authors evaluate their method also for different species with varying evolutionary distances?

From a practical point of view, it would be nice if the authors could address how realistic the method is for non-model species. Could the authors suggest rules-of-thumb for the minimal requirements for the method to be applicable?

2. Does the method explicitly account for the sequence similarity of the given candidate pair? It is not clear from the text, but it seems not? Would including this information as an additional feature improve the classification?

3. Concerning the previous point, I think a comparison of the suggested method with the classical reciprocal-best-hit (RBH) is missing. How much do we gain/lose from the new method compared to relying solely on sequence similarity or RBH. What is the correlation between the ES and the similarity/bit scores? I assume some of the FN (low ES scores) have high sequence similarity, is it just the 6% mentioned in the text?

4. As far as I understand, the method is designed to determine the 1:1 orthology relationship for a pair of candidate genes. Could it also be scaled to more than a pair of species? I think this would be critical for the current pan-genomic era. I think it should be further discussed.

5. From a practical point of view, since the model needs to be trained specifically for the species under study, could the authors provide some general measures for the quality of the model? In addition, it is unclear how much the hyperparameters used for learning are applicable for a new dataset. More specifically: how many iterations should be used? What should be the ratio between the positive and negative set etc? Could this process be done automatically as part of the provided code? It currently seems that the method cannot be used out of the box with a new dataset. I’ll be happy for some clarification and think a detailed protocol/tutorial could greatly enhance the applicability of the suggested method for other species.

Minor

1. ES threshold is mentioned in the abstract but defined only much later in the text, please rephrase or better explain.

2. Could the authors better define and detail “a high-resolution transcriptome map”.

3. Are the AUC values reported based on the cross-validation average? Was part of the data used solely for training and another one for testing as commonly done in machine learning approaches?

4. I suggest the authors would use the precision-recall measure as it is less biased for unbalanced datasets (in this case ~4K positive to 1M negative) (https://www.biostat.wisc.edu/~page/rocpr.pdf)

5. What are the correlation values for Fig 1E?

6. The authors suggest that ‘true’ orthopairs with low ES scores exemplify neofunctionalization events. It is not necessarily the case as it could reflect genes active in different environmental conditions / developmental stages etc. Accounting also for the sequence identity could potentially shade more light on this issue.

7. What are the exact datasets used for this study? Which expression datasets were used for each species? Are they publicly available?

8. Figure 2D legend is missing and not referred to from the main text

9. Why do the authors take the non-normalized read count as a feature? Wouldn’t the normalized count remove some of the biases in each dataset? Does it matter?

10. Which version of OrthoFinder was used?

Reviewer #3: In this manuscript, Kasianov et al. present a method to classify orthologs according to the similarity of their expression profiles. For this, they apply machine learning to expression counts and sequence similarity of genes which are already defined as orthologs (including many-to-many) between two species.

While the aim of the manuscript is of interest, I have several major comments:

While the aim of the method is to discover genes with the same function, the concept of function is never defined in the context of this work.

This manuscript takes as fact that expression is a good and sufficient proxy for function (although the ML does also use sequence similarity, this is never discussed), but there is only one reference given to support this, from a study which validated only 8 genes.

These two points affect all sentences with statements about function. Notably, lines 199-200 and 209, functional similarity or difference have not been shown.

There are several statements comparing this method to others in principle, but it is very surprising for a methods paper to present no comparison to results from other methods. This is absolutely needed, with an exploration of sensitivity to potential confounding factors (e.g. genome annotation, evolutionary distance, quantity of RNA-seq data).

The authors cite as a shortcoming of other methods that they need to be applied to species similar enough to match samples, but the only application given is to two land plants. It would be more convincing to show that genes of similar function could be found between a plant and e.g. a yeast or a fly.

If "the algorithm can be applied to Ribo-seq data, proteomic data, etc." please show evidence for this, by running the method on such data and obtaining relevant results.

Similarly, the method would be more convincing if the authors would show application to more than two species.

The use of results from a single orthology method as input is a potential major weakness of the approach. The authors should test the sensitivity to different orthology methods. Moreover, they explain that orthology methods all have shortcomings, but then they use one of them as input, so it isn't clear how this new approach will help.

The training of the machine learning is very limited. On the one hand are orthologs which are assumed to have similar expression (whereas in other parts of the manuscript this is presented as a hypothesis to test). On the other hand are random pairs which are not matched in any way: not by gene length, not by level of expression, not by GC content, etc. It is trivial that with this approach the orthologs will be way more similar than the random pairs.

I was shocked to find that the code on Github is hard coded for the two species studied here, e.g.:

my $currAthNam = $athList[$athIndex];

my $currFescNam = $fescList[$fescIndex];

Why are read counts not normalized? This is going to leave a large effect of gene length, since longer genes will have more reads for the same level of expression.

Why use "an approach similar to cross-validation" rather than cross-validation?

How are alternative transcripts of the same gene treated for read counting?

How is the method sensitive to the specificity (e.g. tissue-specificity) of expression of genes? Is it better or worse at classifying broadly expressed genes?

Several references, when checked, do not say what they are cited to say. This is worrying, as I didn't check all references.

- ref 19 isn't a test of method efficiency, but a general public review.

- ref 20 used XGBoost for "Natural language processing of text from GEO series" in a paper on crowd-sourcing, not expression data.

- ref 21 doesn't have any reference to XGBoost nor to machine learning as far as I could tell (rapid scan and search for terms in the full text).

- refs 11,28 aren't wrongly cited, but being from 2012 and 2016 cannot be used line 253 to show "growing interest".

I would like to see a reference for the statement lines 257-258 that plants are more diverse in morphology than animals, and thus matching of samples is more difficult. The Plant Ontology is certainly much simpler than Uberon (bilaterian animals), and my experience runs contrary to the statement of the authors.

Minor comments:

line 78-79: what justifies the remark about regulatory genes?

This does not prevent the influence of random fluctuations, it mitigates it; to control for random factors, you can fix the seed for random generators: "To prevent the influence of random fluctuations on the algorithm output, we performed 100 independent iterations of the training and ES calculations"

line 178 "neofunctionalisation" is usually defined between paralogs, not between orthologs.

line 230: either there broad similarity of function among orthopairs, or "widespread inconsistency", but not both.

**Have the authors made all data and (if applicable) computational code underlying the findings in their manuscript fully available?**

Reviewer #1: **No: **Code not fully available. See comments to authors.

Reviewer #2: **No: **I could not find what are the exact datasets used for this study. Specifically, which expression datasets were used for each species?

Reviewer #3: Yes

PLOS authors have the option to publish the peer review history of their article (what does this mean?). If published, this will include your full peer review and any attached files.

Reviewer #1: No

Reviewer #2: No

Reviewer #3: No
---

## [Decision Letter · Decision Letter 1]

30 Aug 2022

Dear Dr Penin,

Thank you very much for submitting your manuscript "Interspecies gene classifier based on the analysis of RNA-seq data using machine learning" for consideration at PLOS Computational Biology. As with all papers reviewed by the journal, your manuscript was reviewed by members of the editorial board and by several independent reviewers. The reviewers appreciated the attention to an important topic. Based on the reviews, we are likely to accept this manuscript for publication, providing that you modify the manuscript according to the review recommendations.

Sincerely,

Andre Kahles, PhD

Guest Editor

PLOS Computational Biology

Ilya Ioshikhes

Section Editor

PLOS Computational Biology

[LINK]

Reviewer's Responses to Questions

**Comments to the Authors:**

Reviewer #1: Kasianov et al have done substantial revisions and new experiments:

- They applied a second sequence-based orthology finder (ProteinOrtho).

- The performed the analysis on more pairs of species: Arabidopsis - maize, buckwheat - maize, Arabidopsis - Arabidopsis,

- They applied another ML method (neural networks with custom regularization)

- They introduced a comparison with Euclidean-distance-based method.

The previously missing parts of the code are now included in the repository.

I only have minor issues below that the authors can be trusted to address without another review.

line 68: These processes [...] makes [...] impossible. => These processes [...] make [...] impossible.

line 158: "input for the algorithm", "First vector is the string". Does the ALGORITHM really use the string rather than the numbers?

I suspect not, although the input to the PROGRAM may be a string representation of numbers, naturally.

line 167: grammar

line 258: We also implemented ICML pipeline => We also implemented the ICML pipeline

line 295: we selected the genes with broad an narrow => we selected genes with broad and narrow

Reviewer #3: In this revision the authors have made many changes and replied to all comments. The manuscript is much improved, and the specific aims of the method are now much clearer.

I still have some comments though.

Major comments:

I was unable to run the method, because the executables from C source did not execute on MacOS, and compilation with gcc gave too many error messages. I could probably fix this, but it is the responsability of the authors to please provide software which functions across platforms.

The authors write several times in the replies to reviewers that the input samples should cover a wide range of conditions, and that they should be from species with sufficient similarity, but this is never clearly defined. The authors should provide specific guidance on the conditions under which their method is expected to work.

Related, please remove the claim page 19 that the method removes the limitation of comparing similar species, since similar body plans are needed, and the plants compared are not more distant than the divergence among mammals.

In the text the authors still mention sub and neo-functionalisation several times, whereas their method does not allow to test such outcomes. I suggest removing all such mentions. Similarly, line 66, genes can be retained by not neo or sub-functionalised (e.g. selection for dosage), please remove this.

Minor comments:

The authors write several times "model object" where I think they mean "model species".

line 85 should be "in recent years this approach was replaced by RNA-seq".

line 251 "the orthologization" should be "the ortholog detection".

line 280 "detalization" doesn't exist, do you mean "higher detail"?

line 303, "is" should be "if" at the end of the line.

**Have the authors made all data and (if applicable) computational code underlying the findings in their manuscript fully available?**

Reviewer #1: Yes

Reviewer #3: Yes

PLOS authors have the option to publish the peer review history of their article (what does this mean?). If published, this will include your full peer review and any attached files.

Reviewer #1: **Yes: **Mario Stanke

Reviewer #3: No

Figure Files:

Data Requirements:

Reproducibility:

References:

---

## [Editor Report · Decision Letter 2]

16 Nov 2022

Dear Dr Penin,

We are pleased to inform you that your manuscript 'Interspecific comparison of gene expression profiles using machine learning.' has been provisionally accepted for publication in PLOS Computational Biology.

Best regards,

Ilya Ioshikhes

Section Editor

PLOS Computational Biology

---

## [Editor Report · Acceptance letter]

5 Jan 2023

PCOMPBIOL-D-21-02191R2 

Interspecific comparison of gene expression profiles using machine learning.

Dear Dr Penin,

I am pleased to inform you that your manuscript has been formally accepted for publication in PLOS Computational Biology. Your manuscript is now with our production department and you will be notified of the publication date in due course.

With kind regards,

Bernadett Koltai
